# Hypophosphatasia: A Unique Disorder of Bone Mineralization

**DOI:** 10.3390/ijms22094303

**Published:** 2021-04-21

**Authors:** Juan Miguel Villa-Suárez, Cristina García-Fontana, Francisco Andújar-Vera, Sheila González-Salvatierra, Tomás de Haro-Muñoz, Victoria Contreras-Bolívar, Beatriz García-Fontana, Manuel Muñoz-Torres

**Affiliations:** 1Clinical Analysis Unit, University Hospital Clínico San Cecilio, 18016 Granada, Spain; juanmivsv@gmail.com (J.M.V.-S.); tomas.haro.sspa@juntadeandalucia.es (T.d.H.-M.); 2Instituto de Investigación Biosanitaria de Granada (ibs.GRANADA), 18012 Granada, Spain; franciscoluisandujar@gmail.com (F.A.-V.); sgsalvatierra@ugr.es (S.G.-S.); victoriacontreras_87@hotmail.com (V.C.-B.); mmt@mamuto.es (M.M.-T.); 3Endocrinology and Nutrition Unit, University Hospital Clínico San Cecilio, 18016 Granada, Spain; 4CIBERFES, Institute of Health Carlos III, 18012 Granada, Spain; 5Department of Medicine, University of Granada, 18016 Granada, Spain

**Keywords:** hypophosphatasia, TNSALP, pyridoxal-5′-phosphate, genotype-phenotype, asfotase alfa

## Abstract

Hypophosphatasia (HPP) is a rare genetic disease characterized by a decrease in the activity of tissue non-specific alkaline phosphatase (TNSALP). TNSALP is encoded by the *ALPL* gene, which is abundantly expressed in the skeleton, liver, kidney, and developing teeth. HPP exhibits high clinical variability largely due to the high allelic heterogeneity of the *ALPL* gene. HPP is characterized by multisystemic complications, although the most common clinical manifestations are those that occur in the skeleton, muscles, and teeth. These complications are mainly due to the accumulation of inorganic pyrophosphate (PPi) and pyridoxal-5′-phosphate (PLP). It has been observed that the prevalence of mild forms of the disease is more than 40 times the prevalence of severe forms. Patients with HPP present at least one mutation in the *ALPL* gene. However, it is known that there are other causes that lead to decreased alkaline phosphatase (ALP) levels without mutations in the *ALPL* gene. Although the phenotype can be correlated with the genotype in HPP, the prediction of the phenotype from the genotype cannot be made with complete certainty. The availability of a specific enzyme replacement therapy for HPP undoubtedly represents an advance in therapeutic strategy, especially in severe forms of the disease in pediatric patients.

## 1. Introduction and Background

Hypophosphatasia (HPP) is a rare genetic disease caused by a loss of function mutation in the gene that encodes tissue non-specific alkaline phosphatase (TNSALP) [1,2]. HPP is considered an innate error of metabolism that presents great clinical variability, but is usually involved in defects in bone and tooth mineralization [3]. The symptoms of HPP vary widely from the most severe forms of the disorder typically found before birth and in early infancy often leading to death to the mildest forms affecting teeth mineralization in adults. In general, the severity of disease is inversely related to the age at diagnosis (except for benign HPP), perinatal lethal and infantile HPP being the most severe forms. Severe forms are characterized by weakness and softness of the bones, causing skeletal abnormalities. Affected infants are born with short limbs, an abnormally shaped chest, and soft skull bones, as well as epileptic seizures caused by an abnormal metabolism of pyridoxal-5′-phosphate (PLP) [4]. Milder forms can include early loss of primary teeth, softening of the bones, and recurrent fractures in the foot and thigh bones linked to chronic pain. Affected adults may lose their secondary (adult) teeth prematurely and are at an increased risk for joint pain and inflammation [5]. Clinical signs, laboratory findings (persistent low alkaline phosphatase (ALP) levels and high PLP levels), radiograph findings, and genetic studies are necessary to diagnose and classify the severity of HPP [6]. However, due to clinical uncertainties of HPP linked to the common symptomatology of other, more prevalent diseases such as osteoporosis or osteomalacia among others, there will probably be a high proportion of the affected population who are underdiagnosed or wrongly diagnosed with other disorders [7].

## 2. Prevalence of HPP

In a disease as rare as HPP, it is difficult to estimate the real incidence. This, linked to the common underdiagnosis of the disease due to the overlapping of its symptomatology with other more frequent disorders, makes the incidence data speculative. In Canada, the birth prevalence of the most severe forms of HPP was estimated at 1:100,000 in the general population based on pediatric records from the Hospital for Sick Children in Toronto [2,8,9]. In this country, the highest prevalence of HPP has been described in the Mennonite community in Manitoba, where approximately 1 in 25 individuals carry a mutation in the *ALPL* gene [10,11], and 1 in 2500 newborns have severe HPP [2]. However, this high proportion is potentially due to a founder effect and therefore results in a very high prevalence in comparison to the norm subsequent processes al population. In the US, HPP appears to be significantly less prevalent in the population of African descent [2,9] the estimated prevalence of severe forms in North America being 1:100,000 [8]. In Japan, the prevalence of HPP was estimated at 1:900,000 due to the homozygous mutation c.1559delT that represents 41% of the HPP alleles [9]. In 2011, the prevalence of HPP in Europe was estimated at 1:300,000 for severe forms using the base of the number of cases originating from France over 10 years while, for moderate forms, the estimate was 1:6370 based on patients with HPP in 20 European countries, this figure being 47 times higher than the frequency of patients affected with severe HPP [9,12]. In 2019, Garcia-Fontana et al. showed that, in Spain, the estimated prevalence of mild HPP could be double that of previously reported estimates (1/3100 vs. 1/6370), and there could be more than 15,000 potential patients affected with mild forms of HPP underdiagnosed due to a lack of recognition in clinical practice [7]. Mornet et al. (2021) show that three kinds of alleles (severe recessive, severe dominant, and moderate) were found with a prevalence on a similar scale, thereby expecting a similar HPP prevalence everywhere in Europe [13]. Although in a previous study Mornet et al. estimated that the prevalence of moderate HPP in the European population was 1/6370 [9], after performing a genetic review of 424 patients with HPP and thanks to additional functional tests, they estimated a higher prevalence of mild forms of HPP (1/2430) [13] than previously reported, similar to prevalence estimated by García-Fontana et al. in 2019 [7].

## 3. The Enzyme

The alkaline phosphatases form a large family of dimeric enzymes common to all organisms [2,14]. They catalyze the hydrolysis of phosphomonoesters [15] with the release of inorganic phosphate.

In humans, there are four isoforms; three of them are tissue-specific (placental isoform (PLAP), the isoform of germ cells (GCAP), and the intestinal isoform (IAP)). They are 90 to 98% homologous in their amino acid sequences, and their genes are clustered on chromosome 2q37.1 [14,16]. The fourth, TNSALP, 50% identical to the other three, is tissue nonspecific and can be found in the bone, liver, and kidney [17,18,19]. Its gene is located on chromosome 1p34–36 [20]. TNSALP is an ectoenzyme that binds to the plasma membrane through a glycosylphosphatidylinositol (GPI) anchor molecule [1,21]. This molecule allows the movement of the enzyme by improving the fluidity of the membrane [2,22].

The extracellular substrates of TNSALP are PLP and inorganic pyrophosphate (PPi) [1,23]. Recent studies also suggest that adenosine triphosphate (ATP), di-phosphoryl lipopolysaccharide (LPS), and phosphorylated osteopontin (p-OPN) are also natural substrates of TNSALP [23,24,25].

TNSALP is a metalloenzyme whose active site contains two Zn^2+^ ions and one Mg^2+^ ion essential for enzyme activity. Furthermore, these metal ions contribute substantially to the conformation of the TNSALP monomer as well as to the indirect regulation of subunit–subunit interactions [16,21]. For the catalytic activity of the enzyme, the homodimerization of the TNSALP monomers is required [2].

## 4. Genetics of HPP

The *ALPL* gene, responsible for HPP, has been shown to be subject to high allelic heterogeneity [1], and more than 400 different mutations have been described, mostly missense mutations (about 74%) [26]. Due to this great variety of mutations, HPP has a highly variable clinical expression [14,21]. This expression ranges from fetal death due to bone hypomineralization to multiple fractures that occur in adulthood [14].

The expression of HPP can be inherited in an autosomal dominant or recessive manner. The more severe clinical phenotypes (perinatal lethal/infantile) are transmitted as an autosomal recessive trait, while the milder forms can be caused by an autosomal recessive or dominant transmission [26]. The detection of a single heterozygous mutation in a patient with mild phenotype HPP with autosomal dominant inheritance can be explained by intronic mutations or mutations in the regulatory sequence or by a heterozygous mutation that has a dominant negative effect (DNE) [27]. Mornet’s group estimated that 13.4% of the HPP-affected chromosome alleles have a DNE in the European population [9]. The autosomal dominant inheritance cases of HPP can be explained by the decrease in wild-type monomer activity in the heterodimeric enzyme complex or by the sequestration and/or degradation of the wild-type monomer in the Golgi apparatus [27].

A DNE modifies the structural and functional properties of the TNSALP homodimer, compromising the kinetic properties of the entire dimer and leading to TNSALP insufficiency and HPP inheritance from generation to generation [23]. The DNE of the in vivo overexpression of TNSALP increases skeletal mineralization and affects the phosphorylation status of osteopontin mutations and can be tested by co-transfection of mutated and wild-type cDNAs and by subsequent measurement of ALP activity [27].

## 5. TNSALP Structure

The analysis of the 3D model of TNSALP based on PLAP showed that the protein is composed of five regions [14]: the active site, which includes three metal-binding sites occupied by two Zn^2+^ and one Mg^2+^ [28,29]; the homodimer binding region, which displays a strong hydrophobic property that dictates the homodimeric structure and thus is crucial for allostery [23,30]; the crown domain, involved in noncompetitive inhibition, heat-stability, allosteric behavior, dimer stability, and collagen binding involved in the mineralization of the extracellular matrix (ECM) [31,32,33]; the calcium binding domain with an unknown function to date [34]; and the N-terminal alpha helix, which contributes to stabilizing the dimeric structure [35] (Figure 1). These regions comprising 78% of the mutations, most of them being severe alleles, clearly indicate that they are crucial for enzyme function and bone mineralization. Mutations in other regions of the molecule are preferentially associated with moderate phenotypes [12,36,37].

Most of the mutations that were shown to have an experimentally measurable DNE were located in three domains: the homodimer binding region, the active site, and the crown domain [27]. Therefore, a mutation located outside of these domains will probably not present a DNE [12]. The mutations that occur in the active center valley of the protein normally lead to less severe phenotypes compared to those located in the rest of the protein regions [38].

## 6. Clinical Manifestations

Patients with HPP have a generalized TNSALP deficiency that is associated with a rare form of rickets and osteomalacia, i.e., poorly mineralized cartilage or bone, respectively [8,16,39]. Although increased fragility fracture rates and recurrent bone marrow lesions are considered a hallmark of HPP due to defective bone mineralization in affected patients [40,41,42], this does not apply to all patients [43] due to the wide variety of genetic mutations. Thus, there are HPP-affected patients showing elevated lumbar spine values in dual X-ray absorptiometry (DXA) and an increased risk for HPP-related fractures [43]. This could largely be explained by a compromised mineralization of the bone microarchitecture. However, to date there are few studies evaluating differential bone involvement between trabecular and cortical bone in HPP patients [44].

Despite the fact that the most common clinical manifestations occur at the bone level, HPP has a very varied clinical expression. The phenotypes range from the complete absence of bone mineralization and fetal death mainly due to respiratory problems associated with thoracic deformities and pulmonary hypoplasia [26,45] to spontaneous fractures, premature tooth loss, seizures, or even nephrocalcinosis [46]. Seizures can begin soon after birth in patients with perinatal lethal or infantile HPP. However, seizures have not been reported to occur in benign prenatal, childhood, adult HPP, or odontohypophosphatasia [47,48,49].

Moreover, HPP patients can present developmental problems as well as multiple complications that affect different organs and systems including muscular, respiratory, renal, and rheumatologic-like symptoms [38] associated with increased articular calcium pyrophosphate deposition (CPPD) [50]. Recent studies have shown that TNSALP may have a regulatory role in bone and muscle progenitor cells influencing musculoskeletal health. Thus, low levels of this enzyme have been associated with muscle weakness and diminished motor coordination [51]. Additionally, it has been revealed that TNSALP can modulate mitochondrial function and ATP levels, and defects in it can be associated with mitochondrial dysfunction, cell respiration, and increased reactive oxygen species, leading to metabolic disturbances in this population [51]. The different manifestations can be classified from the most severe to the mildest forms in perinatal lethal hypophosphatasia, infantile hypophosphatasia, childhood-onset hypophosphatasia, adult hypophosphatasia, odontohypophosphatasia, and benign perinatal hypophosphatasia [8,21,26].

## 7. Subtypes of HPP

### 7.1. Perinatal Lethal Hypophosphatasia

Perinatal lethal HPP is the most severe form and is expressed in utero or at birth with an almost total absence of bone mineralization and can cause fetal death [52]. Radiographic findings are highly indicative of the disease: short-limbed dwarfism, bowed bones, skin-covered osteochondral spurs protruding from the legs or arms (pathognomonic trait), hypoplastic lungs, and defective skull and spine mineralization [2,53,54].

Neonatal death is common in the perinatal lethal form in the first days or weeks after birth [45,55]. Severe respiratory problems due to thoracic deformities and pulmonary hypoplasia are usually the direct cause of around one-half of the deaths in these patients [45,56,57]. The common characteristics of perinatal lethal HPP are the inability to gain weight, irritability, high-pitched crying, periodic apnea, myelophthisic anemia, intracranial hemorrhage, and seizures [21]. These patients usually present with hypercalcemia and hyperphosphatemia that rapidly progress to nephrocalcinosis and kidney damage [58]. As perinatal lethal HPP can be a fatal condition and can be inherited in an autosomal recessive or dominant pattern, prenatal diagnosis is necessary. Furthermore, it is important to perform a careful analysis combining X-rays and the determination of biochemical and molecular markers in order to discriminate benign prenatal HPP from a lethal form of the disease [3,52,59].

### 7.2. ”Benign Prenatal” Hypophosphatasia

In the “benign prenatal” form, a gestational ultrasound shows signs of hypomineralization in the bones of the extremities. These patients present with shortening and bowing of the limbs and, often, dimples that cover deformities of the long bone [60]. However, in an evaluation of 17 patients with benign prenatal HPP, it was revealed that intrauterine deformity can improve spontaneously during the third trimester of pregnancy and may develop normally in some cases [52].

### 7.3. Infantile Hypophosphatasia

The clinical manifestations of infantile HPP occur during the first 6 months of age [2,61]. Craniosynostosis is a manifestation that may appear in infantile HPP and can cause intracranial hypertension, papilemma, or plagiocephaly. On the other hand, a wide fontanelle that slowly closes [48], metaphyseal abnormalities, shaft invasion, and very fragile bones could also appear [61,62]. Other manifestations could be proptosis, mild hypertelorism, and brachycephaly [48]. Due to low bone mineralization, and unlike most types of rickets/osteomalacia, common metabolic abnormalities such as hypercalcemia, hyperphosphatemia, low levels of parathyroid hormone (PTH), and elevated hypercalciuria can occur causing nephrocalcinosis and kidney involvement [48,63].

Progressive skeletal demineralization is accompanied by fractures and bone deformities [63], including bone pain, bowing of the legs, and enlarged joints similar to their occurrence in rickets [3]. Radiolucent “tongues” can also be seen at the extremities of the long bones projecting from the growth plates toward the metaphyses [3]. This form can also be accompanied by respiratory complications due to ricket-like deformities of the thorax [1] and pyridoxine-dependent seizures [23,64]. Pneumonia is a severe complication that can be caused by progressive deformity of the thorax, rib fractures, and tracheomalacia [2]. Both respiratory complications and pyridoxine-dependent seizures are good predictors of lethality in infantile HPP [2,26].

### 7.4. Childhood-Onset Hypophosphatasia

Childhood-onset HPP manifests itself after 6 months or the first year of life, with an especially variable clinical expression from mild to severe [2,26]. One of the most obvious signs that can lead to the diagnosis of HPP is the early loss of primary teeth with intact roots (in the absence of obvious trauma), before the age of three, as it is very unusual in children [62,65]. This is due to the mineralization defect and the aplasia, hypoplasia, or dysplasia of the dental cement that connects the root of the tooth with the periodontal ligament (cementum and alveolar bone) [21,66]. Large pulp chambers and enlarged root canals can also be seen, causing “shell teeth” [67]. 

Skeletal manifestations such as rickets cause short stature, and skeletal deformities may include bowed legs; a dolichocephalic skull with a “beaten copper” appearance; and enlargement of the wrists, knees, and ankles as a result of a widened metaphysis [21,66]. Focal bone defects near the ends of the major long bones can be seen on radiographs, as well as radiolucent “tongues” in major long bones and marked hypomineralization of the metaphyses of the distal ulna and proximal fibula that may guide the diagnosis of HPP [68]. Proptosis, elevated intracranial pressure, and brain damage caused by craniosynostosis may also occur [48].

There are some documented cases of childhood-onset HPP in which chronic recurrent multifocal osteomyelitis (CRMO) occurs, possibly due to spinal edema secondary to the deposition of pyrophosphate crystals [68,69]. Some studies have shown that TNSALP is an anti-inflammatory nucleotidase in mesenchymal cells and neutrophils that acts by dephosphorylating ATP and LPS, and that its deficiency in neutrophils is likely to result in the pathological activation of IL-1β and CRMO [23,25,70,71].

In patients with severe HPP, cases of deafness of unknown origin and in the absence of brain damage have also been documented. Hearing impairment does not occur at birth and can develop over time. Growth hormone deficiency has also been identified in some patients [58] as well as hyperprostaglandinism that can affect the severity of the symptoms [72]. 

Although the clinical manifestations of childhood-onset HPP are usually persistent during growth [67], the spontaneous remission of skeletal manifestations has been described to occur during adolescent life, perhaps due to fusion of the growth plates [2]. The development of the permanent dentition is usually normal, but a resurgence of both dental problems and skeletal disease with osteomalacia can occur in adulthood [1].

### 7.5. Adult Hypophosphatasia

Adult HPP usually occurs in middle-aged adults, around the age of 50 [3,40,73,74]. The loss of adult dentition is a common manifestation in these patients [40,73]. However, some of these patients have suffered the loss of one of their deciduous teeth with complete roots during childhood; some even have a mild history of rickets [40,73]. The first clinical manifestations are usually foot pain due to recurrent metatarsal stress fractures that do not heal [2,23,73] and the deposition of CPPD which causes attacks of arthritis (pseudogout) and pyrophosphate arthropathy [21,75], as well as the ossification of the ligaments (syndesmophytes), similar to what occurs in Forestier’s disease [2] or calcific periarthritis. In calcific periarthritis, a deposition of hydroxyapatite crystals may appear paradoxically near the joints [76].

Pain in the hip can also appear, often causing pseudofractures or complete fractures of the femur [73,77]. Unlike the femoral pseudofracture located in the femoral neck that occurs in osteomalacia, femoral pseudofractures (Looser’s zone) in adults with HPP usually occur in the proximal and lateral subtrochanteric region [77,78].

In addition, these patients can develop other serious complications: chondrocalcinosis, joint inflammation, osteoarthropathy, vertebral fractures, musculoskeletal discomfort, muscle fatigue, hypercalcemia, nephrocalcinosis, and kidney failure [4,40,79]. 

HPP in adults usually has mild symptoms. However, it can become very disabling due to recurring fractures, skeletal and joint pain, and muscle weakness.

Although the most prevalent symptoms in adult HPP are bone, dental, and muscular problems, it has been observed that neurological symptoms can also appear in these patients with a high prevalence. The most prevalent neurological symptoms are fatigue, headache, sleep disturbances, gait changes, vertigo, depression, neuropathy, anxiety disorder, and hearing loss [47].

### 7.6. Odontohypophosphatasia

Odontohypophosphatasia is the mildest and probably most common form of hypophosphatasia [26]. The low activity of TNSALP in HPP results in a reduced absorption of calcium and phosphate [80]. Pyrophosphate dysregulation causes abnormalities that affect the formation of the cementum that covers the root of the tooth, which is essential for tooth structure and fixation to the alveolar bone [81], thus contributing to early dental exfoliation. This form is mainly characterized by dental symptoms with premature spontaneous loss of one or more decidual teeth (typically, the lower and later upper incisors are detached); it occurs painlessly without blood and with the tooth root intact, and enlarged pulp spaces and root canals (“shell teeth”) as the characteristic mineralized cement deficiency impairs the attachment of the tooth root to the periodontal ligament [82]. 

The orodental repercussions of HPP can appear at any age and in all forms of disease, in both infants and children, with the loss of primary teeth, and in adults, with the early loss of permanent teeth and sometimes periodontal disease [62]. In children with HPP, hypercalcemia can lead to poor appetite, eating problems, and severe tooth decay [1]. 

Other dental abnormalities may include bulky crown shape; amelogenesis; and defective dentin, the hard tissue found within enamel [66,80]. Deviations of the root anatomy (taurodontism), defects in the quantity and quality of root cement, short roots, and enamel abnormalities have also been described [80].

## 8. Diagnosis of HPP

Often, HPP patients are underdiagnosed or have a misdiagnosis, with years of HPP evolution until diagnosis [83]. This is because the common symptoms of this metabolic disorder, such as arthralgia, myalgia, and bone pain, are symptoms that are very similar to those that appear in rickets, imperfect osteogenesis, Paget’s disease, rheumatoid arthritis, or fibromyalgia [22]: all of them with a higher prevalence than HPP. Therefore, a strong suspicion and great knowledge of the disease are necessary for diagnosis [84]. Another limiting factor for diagnosis is the low value assigned to a decrease in serum ALP activity, in contrast to the assessment that is usually made for an increase in the activity of this protein [55,84].

Clinical signs and/or symptoms are the first step for the diagnostic suspicion of HPP. However, there are clinical forms during the infant–adolescent period that can present with little or no clinical expression. In these cases, persistently low levels of serum ALP activity could indicate the existence of paucisymptomatic HPP, presenting greater clinical expressiveness in adults. Patients who present signs and/or symptoms at the musculoskeletal and/or dental level should be considered to have a suspected clinical diagnosis of HPP. Respiratory and neurological symptoms that appear in the perinatal lethal and infant forms will strongly support the diagnostic suspicion [85].

The next step after clinical suspicion for the diagnosis of HPP should be the determination of ALP in the serum. Data from normal subjects show that bone ALP contributes approximately half of the total ALP activity in adults [86]. ALP activity that is persistently lower than the reference values may be associated with HPP. However, there are other clinical situations in which the activity of this enzyme is also decreased [86]. Knowledge of these secondary causes will be of great relevance when it comes to a correct diagnosis of the disease [85].

In addition to the more well-known causes of secondary hypophosphatemia, there are other less common situations that are also associated with low levels of ALP, and these must be considered to make an accurate diagnosis. Recent studies have shown that iron and ferritin are potent inhibitors of osteogenesis, significantly inhibiting ALP activity. This is attributed to the ferroxidase activity of ferritin as the central element of this inhibition. Consequently, hemochromatosis should be considered to be a secondary cause, different from HPP, leading to a decrease in ALP levels [87,88] (Figure 2).

On the other hand, a few cases of patients affected by cleidocranial dysplasia, which can mimic HPP, have been described [89]. Cleidocranial dysplasia is a skeletal dysplasia caused by mutations in the bone/cartilage-specific osteoblast transcription factor RUNX2 gene. Although there are clinical manifestations that differ between both pathologies, these patients could present some of the radiographic and biochemical features of HPP. It is characterized by macrocephaly with persistently open sutures, absent or hypoplastic clavicles, dental anomalies, and delayed ossification of the pubic bones. Studies with RUNX2 knockout mice have shown a complete absence of ALP, suggesting that RUNX2 could participate in the regulation of TNSALP activity [90]. However, to date, it is not known whether this is a direct or indirect relation because there are no studies to identify the molecular mechanisms by which RUNX2 and TNSALP interact. 

Additionally, mention should be made of the cases in which the clinical, radiological, and biochemical findings are typical of patients with infantile HPP, with the exception that serum ALP activity is normal or even increased [91,92]. This extremely rare phenotype is called “pseudohypophosphatasia” and has only been convincingly documented in two babies [46].

A possible explanation for these normal levels of ALP is the transient correction of hypophosphatasemia as a result of fractures or intercurrent diseases, or slight elevations in the levels of substrates of the enzyme that causes an overexpression of TNSALP [93,94].

Images compatible with osteopenia, chondrocalcinosis, and/or pathological fractures can be observed after carrying out radiological evaluation in patients affected by HPP. However, these are not pathognomonic findings for HPP [86]. The measurement of bone mineral density by Dual X Absorciometry (DXA) will support the diagnosis of HPP, but without being essential for it [85] since there are HPP-affected patients with a high risk of fragility fractures despite increased lumbar bone density determined by DXA [43]. In this context, it would be very useful to have studies evaluating bone architecture in patients with HPP to assess the contribution of trabecular and cortical bone to the risk of fractures in this population.

The clinical manifestations of HPP are due to the low activity of ALP, which leads to the accumulation of its natural substrates. These substrates are mainly PLP and PPi [23,55]. However, the best candidate for the study of enzyme functionality is PLP, due to the low accessibility for the determination of the substrate PPi. In children and adults affected by HPP, increased plasma levels of PLP are the biochemical marker that best correlates with the severity of the disease, this being the biochemical marker with the highest diagnostic sensitivity [2]. The positive and negative predictive value of PLP for the diagnosis of HPP in childhood is very high, although the absence of elevation of plasma PLP, or other substrates of TNSALP, cannot exclude the diagnosis of HPP [85].

Serum calcium and phosphate and urinary calcium values can also be elevated frequently in early forms of the disease. In general, the levels of PTH and vitamin D are within normal ranges [86].

The finding of pathogenic mutations in the *ALPL* gene allows the definitive diagnosis of HPP to be established. However, the diagnosis of HPP cannot be excluded by not finding a mutation in the *ALPL* gene. This is because the genetic study of *ALPL* is limited to the sequencing of coding regions of the gene (exons), without taking into account the promoter regions or intronic regions [7].

Furthermore, the regulation of the extracellular levels of PPi, responsible for some symptoms of HPP, is complex and involves several genes in addition to the *ALPL* gene [95]. Other studies have reported that there could be mutations in noncoding regions, such as the promoter or intergenic regions of the *ALPL* gene, which are not detected by Sanger sequencing. This could also explain the absence of mutations in this gene in patients with clinical and biochemistry compatible with HPP [96]. In this sense, whole *ALPL* gene sequencing, including the noncoding regions in patients of unclear cases, could be very useful to know if the patients are HPP-affected.

There are other authors who postulate that in the face of a clear clinical suspicion (phenotype + radiology + laboratory findings), a genetic study is not essential for the diagnosis of HPP. However, this information is crucial to document hereditary patterns and the risk of recurrence, as well as for prenatal diagnosis [26,52].

## 9. Genotype–Phenotype Correlations

Several studies have confirmed the great variability in residual enzyme activities of nonsense mutations and the large number of compound heterozygous genotypes that explain the highly variable clinical expressivity [97,98]. For this, transfection studies [1], computer-aided modelling [14,37], and studies of the biochemical properties of ALP in cultured dermal fibroblasts of patients [99] or transfected COS-1 cells [100] were carried out, which allowed researchers to distinguish the severe and recessive alleles from the moderate and dominant alleles [44,98]. While the phenotype in patients with perinatal lethal and infantile HPP is characterized by severe deficits of bone mineralization, rickets, and nephrocalcinosis, HPP in adults with moderate or mild clinical manifestations such as stress fractures with fracture healing disorders, periodontitis, early loss of deciduous teeth, and articular CPPD [44,101]. Severe phenotypes of HPP usually manifest as a recessive disease, produced by homozygous or compound heterozygous mutations [102], while the mild or moderate forms of HPP usually present with DNE of heterozygous mutations [12,27].

The genotype–phenotype correlation may be also studied by observing the phenotypes resulting from recurrent genotypes. In a study by Mornet, 64% of the recurrent genotypes lead to a similar phenotype and 26% led to a phenotype differing from one phenotypic class (e.g., infantile/childhood) [12].

Seizures usually appear in the most severe phenotypes of HPP that have practically no TNSALP activity. The low activity of mutant TNSALP to dephosphorylate PLP on the surface of neuronal cells seems to be related [36]. Furthermore, it was shown that depending on the mutation, the ability of mutant TNSALP to dephosphorylate substrates may differ from one substrate to another (PLP, PPi) [100]. Indeed, Mornet observed that missense mutations in seizure patients appear to be located in particular regions of TNSALP, especially the active site and the calcium site [36].

A recent study by Mornet et al. (2021) classified HPP into three subtypes distinguishable by their genetic characteristics and prevalence: severe HPP (1/300,000), mostly due to homozygosity or compound heterozygosity for severe variants; moderate HPP (1/2,430), mainly due to the DNE of missense variants; and mild adult HPP, characterized by unspecific signs (1/508). Mornet hypothesizes that mild HPP is possibly due to a haploinsufficiency mechanism with a negative interaction between TNSALP and another actor, possibly the collagen matrix [13].

In cases where adult HPP is not obvious, due to inconclusive genetic findings associate to rare genetic variants of the *ALPL* gene or to genetic variants of uncertain pathogenic significance (American College of Medical Genetics and Genomics—ACMG class 4) could be difficult to diagnose the disease. In those patients, HPP should only be diagnosed when persistently decreased TNSALP activity and elevated PLP levels are shown, in addition to skeletal manifestations and at least one additional complication related to the teeth, the musculoskeletal system, or the central nervous system [97]. In these patients, it is of great interest to carry out in vitro studies to determine the percentage of loss of enzymatic activity associated with a mutation of uncertain significance to determine whether the mutation is pathogenic or not [7]. 

Currently, more than 400 different mutations have been described in the *ALPL* gene and annotated in the *ALPL* mutation database (http://alplmutationdatabase.hypophosphatasie.com/, accessed on 11 January 2021). 

The correct identification of new variants and the study of their phenotype will allow the improvement of knowledge of this metabolic disorder to make it accessible for the scientific community, allowing a better management of disease.

## 10. Treatment of HPP

The approaching of HPP must be multidisciplinary and individualized based on the clinical manifestations being the treatment of related symptoms, the measure most used [84]. In perinatal lethal HPP, the patient may develop pulmonary hypoplasia, requiring respiratory support to control respiratory failure [55,63].

Different therapeutic approaches with disparate results are described in the literature. Bone marrow transplantation in infants has been tried with satisfactory results [103,104], while growth hormone therapy in children with HPP has also shown a significant increase in ALP activity and increased growth [105,106].

The clinical manifestations secondary to the deposition of calcium pyrophosphate or hydroxyapatite crystals respond to non-steroidal anti-inflammatory drugs or to glucocorticoids locally [84]. In addition, the use of cycles of non-steroidal anti-inflammatory drugs gives good results for persistent pain secondary to fractures [72,107].

Another therapeutic approach would be the use of PTH or monoclonal anti-sclerostin antibodies. PTH’s mechanism of action is based on the direct stimulation of bone formation [108]. Isolated cases of HPP treated with teriparatide (PTH 1–34 or PTH 1–84) at low doses have been described as inducing with a “paradoxical reaction” [85], with decreased pain, better radiological consolidation of pseudofractures or stress fractures [109,110,111], and an improvement in biochemical and densitometric parameters [112]. However, cases have also been described in which the response to teriparatide treatment may vary depending on the TNSALP mutation [113,114,115]. An increase in ALP values, a decrease in PLP concentration, and an increase in lumbar bone mineral density (BMD) have also been observed with teriparatide treatment, with stability of femoral BMD [112]. Monoclonal anti-sclerostin antibody treatment has been demonstrated to increase bone formation and reduce bone resorption, increasing the BMD in HPP patients [116].

The use of calcium and/or vitamin D3 supplementation is not recommended in HPP-affected patients with normal calcium levels because it could cause hypercalcemia, hyperphosphatemia, and hypercalciuria [85]. 

It is important to note that bisphosphonate treatment in adults with HPP may increase fracture risk [74,117,118], since these are analogues of PPi and therefore contributing to the inhibition of hydroxyapatite formation.

Several enzyme replacements therapies have been evaluated to treat HPP. Among them, the use of the transfusion of blood plasma rich in TNSALP has been used. However, biochemical, clinical, and radiological improvement was only observed in some patients, but transiently and with nonreproducible results. This appears to be because the treatment is effective only if the TNSALP is incorporated into the bone structure [85]. 

The enzyme replacement therapy that has shown the greatest effectiveness to date is recombinant human TNSALP (asfotase alfa). It is a protein formed by the catalytic domain of TNSALP, the Fc fragment of IgG1, and a deca-aspartate motif that increases the affinity for its binding to hydroxyapatite [63]. The first results of the therapeutic study with asfotase alfa in 2012 in patients with severe HPP of the perinatal lethal or infantile forms demonstrated a significant improvement in skeletal, muscular, pulmonary, and cognitive and motor development. A rapid decrease in plasma levels of PLP and PPi was also observed [63]. However, some therapy-related adverse events have been reported, the most frequent being a local reaction at the subcutaneous injection site. Likewise, other side effects such as kidney, cornea, and conjunctiva calcifications have been reported in up to 55% of the juvenile-onset cases, without any change reported in vision or in renal function [5].

Recent studies have shown the effectiveness of asfotase alfa treatment in adults and adolescents with pediatric-onset HPP, showing an improvement of osseous consolidation of non-unions [119], a normalization of circulating TNSALP substrate levels, and improved functional abilities and health-related quality of life [119,120,121], without severe adverse events. 

Treatment with asfotase alfa requires the correct therapeutic monitoring, both to adjust the dose and to control possible adverse effects. In addition, the appearance of ectopic mineralization should be monitored, maintaining radiological, renal ultrasound, and ophthalmological controls [122].

## 11. Conclusions

Although HPP is mainly due to one or more mutations in the *ALPL* gene leading to a decrease in TNSALP activity and the accumulation of natural substrates of the enzyme, there are other situations that can lead to low levels of serum ALP without any mutations in the *ALPL* gene. These situations could include mutations in noncoding regions of the *ALPL* gene; defects in transcription factors such as RUNX2, which is involved in the regulation of TNSALP and therefore in the regulation of bone mineralization, presenting phenotypes that may overlap with those of HPP; or other secondary causes unconsidered to date such as hemochromatosis. 

Despite the difficulty of correlating genotype with phenotype, there is the possibility of linking the phenotype by studying recurrent genotypes. The influence of other genetic, epigenetic, or non-genetic factors for HPP could explain the cases in which there is greater difficulty in relating the genotype to the phenotype.

An adequate detection protocol is crucial for diagnosing this type of metabolic disorder, considering the persistent low serum levels of ALP in patients, in order to further the molecular and genetic study to diagnose this disease in the early stages and to avoid the worsening of symptoms due to inappropriate treatments such as bisphosphonates. 

The accurate diagnosis of HPP is of great importance due to the potential severity of the disease, either due to the marked impairment of the quality of life associated with the disease or the possible iatrogenesis derived from an erroneous diagnosis. All of this coupled with the recent availability of a specific enzyme replacement therapy for HPP (recombinant human TNSALP, asfotase alfa), especially in severe forms in the infant–juvenile period, makes a correct diagnosis necessary for early and adequate treatment. This enzyme replacement therapy is being studied in adult patients and is currently showing promising results. However, more studies are necessary because this therapy seems to have some adverse reactions, e.g., vascular calcifications reported in a high proportion of juvenile patients treated. 

## Figures and Tables

**Figure 1 ijms-22-04303-f001:**
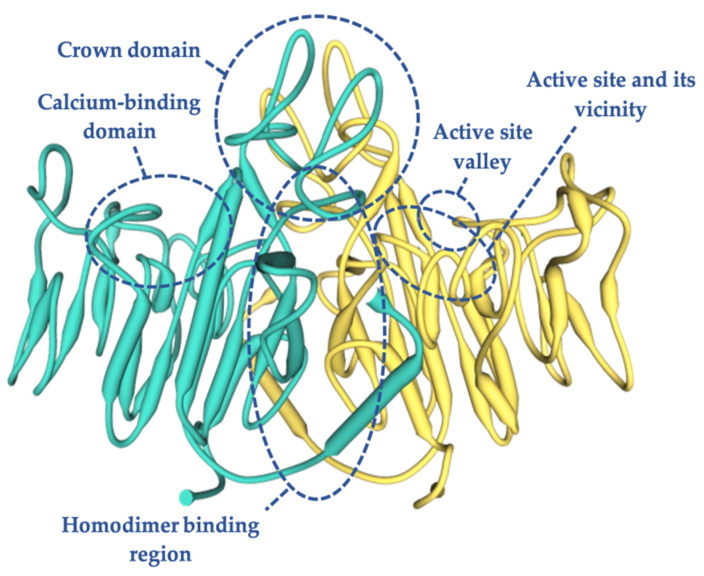
Three-dimensional structure of human TNSALP based on the placental isoform of the ALP structure (~74% homology), obtained using a simulation model of the SWISS MODEL program by means of an amino acid sequence obtained from UniProtKB (P05186-PPBT_HUMAN). In green and yellow each monomer is shown.

**Figure 2 ijms-22-04303-f002:**
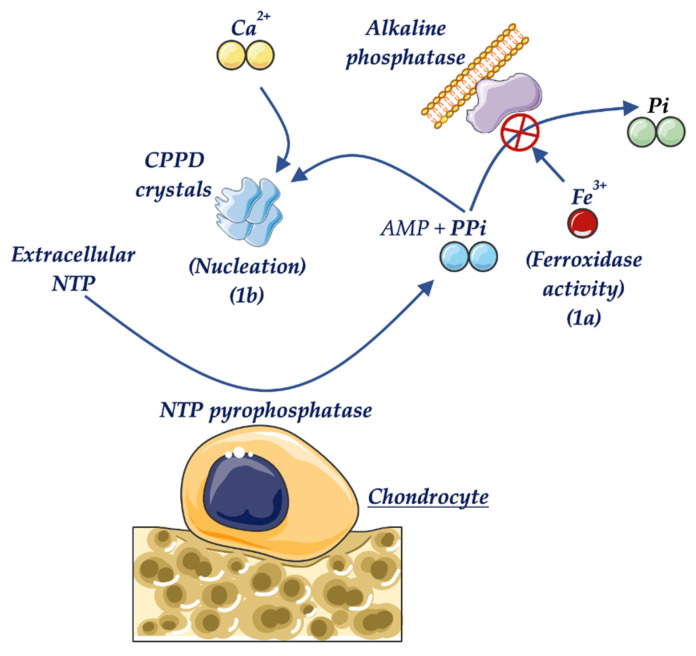
Metabolism of (PPi) and formation of CPPD crystals in hemochromatosis. (1a) Inhibition of ALP by iron, resulting in high PPi values. (1b) The nucleation activity of iron in CPPD crystal formation.

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
