# Peer review of "Hypophosphatasia: A Unique Disorder of Bone Mineralization"

_ijms, 2021, doi:10.3390/ijms22094303_

Round 1

Reviewer 1 Report

  • I do not understand this sentence in the abstract “the prevalence of severe forms is more than 40 times the prevalence of mild forms”. Authors should indicate in a sentence how a severe and a mild form is distinguished from one another? The same remark can be done in chapter 2 about the two references from Etienne Mornet. Which criteria was used to separate severe from mild forms? As written in the last sentence of the paragraph, mild forms are moreover often underdiagnosed.
  • Chapter 6: there is a shift in the references, with for instance Zhang Z et al (Int J Mol Sci 2021) being cited as ref 40 or 41 in the text but being ref 39 in the reference list.
  • Chapter 7, when discussing the cases of recurrent chronic multifocal osteomyelitis in children, authors should mention the emerging role of alkaline phosphatase as an anti-inflammatory enzyme, acting by dephosphorylating ATP and LPS.
  • Chapter 7.7 title should be written in English

Author Response

Response to Reviewer 1 Comments

First, we want to thank you for your comments, which have undoubtedly contributed to improve the quality of our manuscript. Please, find below the responses to your kind suggestions:

Point 1: I do not understand this sentence in the abstract “the prevalence of severe forms is more than 40 times the prevalence of mild forms”. Authors should indicate in a sentence how a severe and a mild form is distinguished from one another? The same remark can be done in chapter 2 about the two references from Etienne Mornet. Which criteria was used to separate severe from mild forms? As written in the last sentence of the paragraph, mild forms are moreover often underdiagnosed.

Response 1: As you point out, there is a mistake in the sentence of abstract. This mistake has been changed by the correct sentence “It has been observed that the prevalence of mild forms of the disease is more than 40 times the prevalence of severe forms”

On the other hand, the chapter 1 (Introduction and background) has been rewritten including the main clinical signs characteristics of severe and milder forms of HPP. This information is explained more detailed in chapter 7 (Subtypes of HPP).

The best protocol to separate severe from mild forms is the analysis of clinical signs, laboratory findings, radiograph findings and genetic study as we have described in chapter 1. We also have point out the problem of underdiagnose of this disease due to unknown of it in clinical practice linked to the overlap of the symptomatology with other more prevalent disorders.

Point 2: Chapter 6: there is a shift in the references, with for instance Zhang Z et al (Int J Mol Sci 2021) being cited as ref 40 or 41 in the text but being ref 39 in the reference list.

Response 2: This problem has been corrected in the new version. All references have been reviewed and updated and now, Zang Z et al fits with the sentence cited in the text (ref 50).

Point 3: Chapter 7, when discussing the cases of recurrent chronic multifocal osteomyelitis in children, authors should mention the emerging role of alkaline phosphatase as an anti-inflammatory enzyme, acting by dephosphorylating ATP and LPS.

Response 3: Thanks for this recommendation. As you indicate, we have added the description of the role of alkaline phosphtase as an anti-inflammatory nucleotidase in the section of Childhood-onset hypophosphatasia as follows:

“Some studies have shown that TNSALP is an anti-inflammatory nucleotidase in mesen-chymal cells and neutrophils that acts by dephosphorylating ATP and LPS, and that its deficiency in neutrophils is likely to result in the pathological activation of IL-1β and CRMO [73]”

Additionally, the following reference has been included to support the explanation:

Graser, S.; Liedtke, D.; Jakob, F. TNAP as a New Player in Chronic Inflammatory Conditions and Metabolism. Int J Mol Sci 2021, 22, doi:10.3390/ijms22020919

Point 4: Chapter 7.7 title should be written in English

Response 4: The translation has been done and this paragraph has been moved to Chapter 8 as recommended by the second reviewer

Reviewer 2 Report

The Review manuscript from Villa-Suárez et al. “Hypophosphatasia: a unique disorder of bone mineralization” summarizes TNSALP function, HPP diagnostics and other clinical information concerning this inherited rare disease. In my opinion, the manuscript summarizes the current literature only to a certain extent and lacks fine-tuning. Accordingly, I have to raise a few points of criticism:

Major points:

- Whole document: Check for correct gene nomenclature. All human gene names, e.g. ALPL, are by definition written in italics. Furthermore, check for correct usage of protein writing or nomenclature of non-human genes/proteins.

line 165, ff: Subtypes of HPP.

1) The classification of different HPP subtypes entitled as “7.3. Childhood hypophosphatasia” and “7.4. Infantile hypophosphatasia” is in my opinion wrong and in its current form misleading.  Either the description of both subtypes have been confused, or are incorrectly used (check for Childhood hypophosphatasia = ORPHA: 247667; Infantile hypophosphatasia = ORPHA: 247651). Please recheck your classification carefully, control the described HPP manifestations for each classification and maybe rewrite the section.

2) Some clinical observations of certain subgroups are not mentioned in full detail (e.g. neurological symptoms, like depression, fatigue, anxiety disorder, in adult HPP patients) and need to be more precise (e.g. cause of death in perinatal form is almost always lethal within days or weeks, and around one half of patients with the infantile form die from respiratory complications.).  

3) Grouping of HPP like cases in  “Pseudohypophosphatasia” (please correct your headline spelling) is not a real clinical HPP subtype by ORPHA definition, but summarizes similar phenotypes in comparison to HPP. Nowadays, the phrase is rather seldomly used in current literature. To further distinguish this you could move this part into an own section and include further statements from your review within, e.g. on RUNX2 mutations.
Orphanet reference numbers to subtypes and comparison to Orphanet description (https://doi.org/10.1186/1750-1172-2-40) might be useful.

- fig. 1/ line 130: Formation symbols of used font are visible. Text is partly missing. Text is hard to read, as it is overlaying the 3D structure. Homodimer description is a misleading/elusive, as the green and yellow parts of the shown structure are the monomers and both monomers are shown. Did you mean “homodimer binding region”?  Please rework figure.

- lines 316, ff, keywords and abstract: The authors describe a suspected link between HPP and Cleidocranial dysplasia (CD; RUNX2 mutations), due to similar symptoms and clinical observations in both diseases. In my opinion, this point is overstated within the review, is partly misleading and should be rephrased, due to several reasons:  

1) Until today only a small number (I could only find three case reports) of patients have been reported and therefore resemble a very small fraction of cases.

2) A molecular link between TNAP and RUNX2 either by cell signaling or by transcription factor activity has been described in both directions and make clear dissection between both gene function very hard (e.g. see Nakamura et al. “Tissue-nonspecific alkaline phosphatase promotes the osteogenic differentiation of osteoprogenitor cells” Biochem Biophys Res Commun. 2020 Apr 9;524(3):702-709. doi: 10.1016/j.bbrc.2020.01.136 and Yoshida et al “SP7 inhibits osteoblast differentiation at a late stage in mice” PLoS One. 2012;7(3):e32364.doi: 10.1371/journal.pone.0032364). RUNX2 function is moreover very prominent in skeletal cells due to differentiation processes and might override a potential reduction of TNAP function secondarily, just by reducing corresponding cell numbers. A direct interaction of TNAP activity and RUNX2 is rather unlikely (as stated in line 323).

3) Clinical manifestation of RUNX2 and ALPL mutations can be rather different, e.g. HPP patients have been reported to show craniosynostosis, while CD patients lack this completely and display persisting open sutures. A second example: lack of mineralization in severe HPP forms is rather general, while loss of clavicle in CD patients is mostly a singular event effecting not all skeletal parts.   

- The review superficially describes neurological symptoms in HPP patients only in some parts. This point should, at least for the HPP subtypes, be enhanced and/or corrected. e.g. seizures are described for severely affected children, while depression is only described in adults. See also Colazo, et al. “Neurological symptoms in Hypophosphatasia” Osteoporos. Int. 2019 Feb;30(2):469-480. doi: 10.1007/s00198-018-4691-6.

  - The manuscript incorporates a number of writing errors in its current form. Most times missing or too much spaces, e.g. after Ref. [1,2] line 47 and line 62 after “Toronto”. Besides these typesetting errors, a number of grammatical errors are obvious, e.g. line 109 “negative dominant” should be “dominant negative”, etc. Please carefully check the manuscript for these and correct them.

Minor points: 

- line 7 to 22: The affiliation “Endocrinology and Nutrition Unit …” has been named three times.

- line 44: HPP is potentially life-threatening for severe cases, but not for all its manifestations. Please rephrase. 

- line 48-49: A fitting reference to this sentence would be nice to add.

- line 52: “…and dental problems in adults caused by the accumulation of inorganic pyrophosphate (PPi)” PPi is only one , but probably not the only reason for dental phenotypes. Please clarify.

- line 57, ff: The section is a bit unorganized (could be enhanced by changing order of sentences) and misses new findings. The Mennonite community described in the first sentence is potentially due to a founder effect and thereby results in a rather very high prevalence in comparison to normal population subsequently described.

- line 74, ff: One prominent feature of TNAP is the location of the enzyme to cell membranes by its GPI anchor (ectoenzyme). This fact is not mentioned in the paragraph.

- line 75: Please add a reference to this sentence.

- line 78-80: Missing reference and please state kind of homology (amino acid, nucleotide, DNA, mRNA).

- line 83: Comment to PEA. Currently the scientific evidence for direct turnover of PEA by TNAP is lacking. PEA is used as a diagnostic readout for TNAP function, but to my knowledge it has not been reported how TNAP and PEA are biochemically reacting as direct binding partners.    

- line 91: Change heading to “Genetics of HPP”

- line 99: “…caused by …” maybe “inherited” is more appropriate.  

- line 106: “..the inhibition of wild-type monomer activity..” A bit misleading, as residual TNAP function is not fully inhibited, but is rather reduced in these patients. To my knowledge, a residual TNAP activity can be detected even in autosomal dominant patients. This observation is also nicely summarized in E. Mornet 2007 https://doi.org/10.1186/1750-1172-2-40.

- line 113: To measure dominant negative effects in vitro, cells have to co-transfected with mutated and wild-type cDNAs and subsequently measurement of TNAP activity has to be conducted. Please be more precise.   

- line 114, ff: Currently there is no 3D model of TNSALP reported. Most times TNAP 3D models based on its homology to PLAP (~74% homology) are displayed. Please clarify this point in your text similar to the statement in your figure 1 legend.

- line 140: Fetal death is not automatically caused by the lack of mineralized structures, but in most cases by respiratory complications, which are only partly depending on bone structure.  

- line 141: "decidous teeth"  or "decidual teeth". Please check correct word use.

- line 141: A bit misleading phrase, as adults do not lose decidual/decidous teeth, but infants do.

- line 156: “…has…” -> low levels…had been

- line 157: Please clarify this statement. Reference 40 describes TNAP function, while it is bound to cellular membranes via its GPI anchor, and not in its soluble/sheeded form. For me Ref 41 is misleading in this context, the corresponding review is mainly on mitochondria but has no direct link to HPP.  

- line 158: “... level sand..” typo, should read “… levels and..” 

- line 286: “… evolution [82].” wrong word ?? Did you mean “evaluation”?

- Comment to line 351: Nowadays, performing of Whole-genome resequencing in unclear diagnosed patients is discussed and can solve this problem.

- line 364, ff: This section should include novel findings and the authors might want to include E. Mornet Paper “Hypophosphatasia: a genetic-based nosology and new insights in genotype-phenotype correlation” European Journal of Human Genetics (2021) 29:289–299.

- line 367: “In addition, these tests allowed…” Unclear. Which tests do you refer to? Please clarify.

- line 374, ff: This statement is very general and in my opinion misleading. Distinct clinical HPP groups have specific modes of inheritance. Especially prenatal and perinatal HPP groups can have severe manifestation, but display different modes of inheritance.

- line 383-384: “Most of the patients with this neuronal phenotype have practically no TNSALP Activity..” The statement is only true for severe cases displaying cerebral seizures, but for other neurological HPP symptoms (e.g. depression, fatigue) reported for adult patients, this statement is wrong and should be rephrased.   

- line 418-422: In my opinion is this section is misplaced and should be moved rather to the end of the chapter before enzyme replacement therapies.

- line 482-484: “This enzyme replacement …” The statement is misleading in its current form, as asfotase alfa treatment is already an authorized treatment for severely affected patients/children and adult forms of HPP are normally not classified as severe.

- line 491, ff: Data Availability Statement is currently not formulated, but the standard journal formulation is left in the manuscript

- line 529, ff/ reference 17: link directly not accessible in its current form. Maybe give reference to the corresponding text book, Print ISBN: 9783527310791.

Summary:

In my opinion, the present manuscript does not completely fulfil its purpose to inform the reader broadly about HPP in a comprise form and is even partly misleading in its current form. The HPP classification cannot be reconstructed and is in my opinion wrong. The authors should carefully check the manuscript for typos, correct gene names, spacing and grammatical errors. 

Round 2

Reviewer 1 Report

Authors have answered my comments. I suggest to include with ref 73, which is a review article , the experimental articles on ATP and/or LPS dephosphorylation by TNAP

Reviewer 2 Report

The Review manuscript from Villa-Suárez et al. “Hypophosphatasia: a unique disorder of bone mineralization” summarizes TNSALP function, HPP diagnostics and other clinical information concerning this inherited rare disease. In my opinion, the authors have revised their manuscript rigorously according to the reviewer comments and the current manuscript to my opinion shows only a few minor points of criticism left:

Minor points: 

-line 50-51: “In general, the severity of disease is inversely related to the age at diagnose, perinatal lethal and infantile HPP being the most severe forms.” – The statement is not fitting to benign HPP form, which can be diagnosed early, but normally is not lethal. “diagnose” should be “diagnosis”   

-line 73: “…gene[10, 11],…” – space is missing.

- Comment to line 87-89: One additional, new aspect of the Mornet paper is that prevalence of the mild HPP form is unexpectedly high.  

- line 150-153: The sentence is highly similar to the given reference and should be rephrased.

- line 495: “… since these are analogues of PPi contributing to the ..” subsequent processes, maybe rephrase to “ … since these are analogues of PPi and therefore contributing to the…”
